# Endoscopic Resection of Residual or Recurrent Lesions after Circumferential Radiofrequency Ablation for Flat Superficial Esophageal Squamous Cell Neoplasias

**DOI:** 10.3390/cancers15143558

**Published:** 2023-07-10

**Authors:** Yung-Kuan Tsou, Chi-Ju Yeh, Puo-Hsien Le, Bo-Huan Chen, Cheng-Hui Lin

**Affiliations:** 1Department of Gastroenterology and Hepatology, Chang Gung Memorial Hospital, Taoyuan 33353, Taiwan; flying@adm.cgmh.org.tw (Y.-K.T.); b9005031@cloud.cgmh.org.tw (P.-H.L.); spring03258@adm.cgmh.org.tw (B.-H.C.); 2Department of Medicine, College of Medicine, Chang Gung University, Taoyuan 33353, Taiwan; t22259@adm.cgmh.org.tw; 3Department of Pathology, Chang Gung Memorial Hospital, Taoyuan 33353, Taiwan

**Keywords:** esophageal squamous cell neoplasia, radiofrequency ablation, endoscopic resection, endoscopic submucosal dissection, residual, recurrent

## Abstract

**Simple Summary:**

We report the efficacy and safety of endoscopic resection of residual/recurrent superficial esophageal squamous cell neoplasias (SESCNs) after circumferential radiofrequency ablation (RFA). SESCN patients treated with primary endoscopic submucosal dissection (ESD) served as a control group. Endoscopic mucosal resection failed to remove one residual SESCN. The pathological results of the 16 resected specimens were classified into three groups: high-grade intraepithelial neoplasia (HGIN) without ductal/submucosal glandular involvement (37.5%), HGIN with ductal/submucosal glandular involvement (25.0%), and cancer with muscularis mucosae or deeper involvement (37.5%). These three groups may imply three possible routes in which residual/recurrent SESCNs occurred. Compared with the control group, the study group had similar procedural speed, en bloc resection rate, R0 resection rate, and complication rate. In conclusion, the safety and efficacy of post-RFA ESD were similar to those of primary ESD. We recommend that ESD should be the treatment of choice for residual/recurrent SESCNs after initial RFA.

**Abstract:**

The optimal treatment of residual/recurrent superficial esophageal squamous cell neoplasias (SESCNs) after circumferential radiofrequency (RFA) remains unclear. We aimed to report the efficacy and safety of endoscopic resection (ER) of residual/recurrent SESCNs after RFA. Patients who underwent circumferential RFA with residual/recurrent SESCNs and were treated with ER were retrospectively collected. SESCN patients treated with primary endoscopic submucosal dissection (ESD) served as the control group. Eleven patients who underwent RFA had a total of 17 residual (*n* = 8) or recurrent (*n* = 9) SESCNs and were treated for ER. EMR failed to remove one residual SESCN. Of the 16 resected specimens, 10 were high-grade intraepithelial neoplasia (HGIN) and six were cancer. Eight cases had neoplasia extending to esophageal ducts/submucosal glands (SMGs). The pathological results may imply three possible routes in which residual/recurrent SESCNs occurred: HGIN without ductal/SMG involvement (37.5%), HGIN with ductal/SMG involvement (25.0%), and SCC with muscularis mucosae or deeper involvement (37.5%). Compared with the control group, the study group had similar procedural speed, en bloc resection rate, R0 resection rate, and complication rate. In conclusion, the safety and efficacy of post-RFA ESD were similar to those of primary ESD. ESD should be the treatment of choice for residual/recurrent SESCNs after initial RFA.

## 1. Introduction

In recent years, superficial esophageal squamous cell neoplasms (SESCNs), including high-grade intraepithelial neoplasia (HGIN) and mucosal squamous cell carcinoma (SCC), have been primarily treated endoscopically [1]. Endoscopic submucosal dissection (ESD) has become the mainstay of endoscopic treatment in SESCNs, but it still has limitations [2,3,4]. SESCNs often arise and expand in a patchy manner along the esophageal surface [5]. Consequently, some SESCNs are extensive (large and/or multiple, or with multifocal Lugol-unstained lesions in the background) [5,6,7,8]. Extensive SESCNs usually require more complex treatment modalities. Despite technical feasibility, ESD for extensive SESCNs can result in postoperative esophageal strictures, leading to the need for multiple endoscopic dilatations, negatively impacting the quality of life, and delaying additional treatments such as chemoradiotherapy. Therefore, guidelines issued by the Japan Gastroenterological Endoscopy Society recommend that ESD of circumferential lesions should be limited to cT1a-m1/m2 SESCNs no longer than 5 cm in length [9]. In this context, radiofrequency ablation (RFA) has recently been reported and may be superior to ESD for early flat SESCNs as it results in fewer major procedure-related adverse events [10,11,12]. However, the main problem with RFA is the lack of post-treatment specimens to assess the curability of the entire lesion. Therefore, some patients may require multiple treatments of RFA to achieve complete remission [13,14,15,16]. In addition, despite repeated RFA treatments, post-RFA residual lesions, local recurrence, and even disease progression had been observed [14,15,17]. Therefore, repeat RFA for residual or recurrent lesions after an initial RFA may not be a good option. Conversely, endoscopic resection (ER), including ESD and endoscopic mucosal resection (EMR), may be a better treatment option for residual or recurrent lesions after initial RFA, but studies are lacking. Therefore, this study aims to investigate the efficacy and safety of ER for residual or recurrent SESCNs after initial circumferential RFA of extensive flat SESCNs.

## 2. Materials and Methods

### 2.1. Patient Selection

Our institution is a tertiary referral center, and since 2012, ESD has been the mainstay of treatment for SESCNs. RFAs for SESCNs began in late 2018, primarily for patients with extensive flat SESCNs who were not candidates for esophagectomy. Extensive SESCN was defined as (1) at least one SESCN invading >3/4 of the esophageal circumference and greater than 5 cm in length (*n* = 6), or (2) multiple Lugol-unstained lesions, with/without connections, ranging from >3/4 esophageal circumference and length greater than 5 cm (*n* = 5). The study flowchart is shown in Figure 1. Between January 2019 and December 2021, 19 patients who underwent RFA for SESCNs were retrospectively identified from the computerized database of our Therapeutic Endoscopic Center. Patient follow-up data were updated in December 2022, or until death. The inclusion criteria were (1) Early SESCNs, defined as biopsy showing HGIN; (2) Flat lesions, defined as Paris classification type 0-IIb or 0-IIb + 0-IIc; (3) Extensive SESCN; (4) Endoscopic ultrasonography (EUS) showing SESCN confined to the mucosal layer; (5) Computed tomography (CT) showed no lymph nodes or distant metastases. The exclusion criteria were: (1) Patients who received focal RFA (*n* = 2); (2) Patients who received ESD combined with RFA at the same time (*n* = 1); (3) Patients without residual or recurrent SESCN after primary RFA during follow-up (*n* = 5). This study was reviewed and approved by the Ethics Committee of the Chang Gung Memorial Hospital (IRB No.: 202300643B0). Since this was a retrospective study using routine treatment or diagnostic medical records, the Chang Gung Medical Foundation Institutional Review Board approved the waiver of the participants’ consent.

### 2.2. Radiofrequency Ablation

RFA was performed using the Barrx™ Radiofrequency Ablation System. In 2019, the HALO 360 ablation system was used, and the regimen was ablation (12 J/cm^2^)-clean-ablation (12 J/cm^2^) (*n* = 2). After 2019, the 360 Express ablation system was used, and the regimen was ablation (10 J/cm^2^)-clean-ablation (10 J/cm^2^) (*n* = 6) or later ablation (12 J/cm^2^)-clean-ablation (10 J/cm^2^) (*n* = 3). The initial RFA received by these patients was defined as the index RFA. The treatment area (TA) was defined as the area from 1 cm proximal to 1 cm distal to the Lugol-voiding lesions-bearing segment of the esophagus. Two endoscopists performed the RFA procedures (Y.-K.T. and P.-H.L.)

After the index RFA, image-enhanced endoscopy (IEE), including narrow-band imaging and Lugol chromoendoscopy, was performed during follow-up. The first IEE was conducted 2–3 months after the index RFA. Residual SESCN was defined as any lesion found within the TA of the index RFA during the first IEE, and pathology revealed HGIN or SCC. If there was no residual SESCN at the first IEE, subsequent IEEs were performed every 3–6 months during the follow-up period. Recurrent SESCN was defined as HGIN or SCC detected within the TA of the index RFA on subsequent IEE. Due to the COVID-19 pandemic during the study period, some patients had their first or subsequent IEE delayed.

### 2.3. Endoscopic Resection

Before ER, CT scans were performed for residual SESCNs, and EUS and CT scans were performed for recurrent SESCNs to restage the disease. Only those SESCNs confined to the mucosa without lymph nodes or distant metastases were eligible for ER. For pathologically proven SCC, either before or after ER, positron emission tomography-CT scans were also performed to assess N and M staging. The EMR method we used was EMR-C [18]. The detailed procedure for ESD was similar to that described in our previous report [19]. Briefly, lesions were identified using Lugol chromoendoscopy. Glycerol mixed with indigo carmine was used for submucosal injection. Unlike previous reports, we used a Dual Knife J (KD-655; Olympus Medical Systems, Tokyo, Japan) or an IT Knife Nano (KD-612L, Olympus Medical Systems, Tokyo, Japan) to perform ESD. The ER procedures were performed by an experienced endoscopist (Y.-K.T.) and two young endoscopists (B.-H.C. and C.H.L.) performing ESD under the supervision of (Y.-K.T.).

### 2.4. Control Group

During the 2019–2022 study period, an additional 87 SESCNs underwent primary ESD without prior RFA treatment. To clarify the effect of ESD on residual or recurrent SESCN after RFA, we selected cases from these 87 cases in the control group whose tumor length was within the range of the tumor length observed in the study. As a result, there were 74 cases in the primary ESD group.

### 2.5. Statistical Analyses

In the text and tables, the data of continuous variables were expressed as median and range, while the data of categorical variables were expressed as a number (%). To compare the results of the post-RFA ESD group and the primary ESD group, the Mann-Whitney U test was used to compare continuous variables, and the chi-square test or Fisher’s exact test was used to compare categorical variables. A two-tailed *p*-value < 0.05 was considered statistically significant. Statistical analysis was performed using SPSS software (Version 22; SPSS, Inc., Chicago, IL, USA).

## 3. Results

A total of 11 patients who met the inclusion and exclusion criteria were included in this study. After the index RFA in these 11 patients, 17 residual or recurrent SESCNs were diagnosed (Figure 2a) and underwent ER.

### 3.1. Index Radiofrequency Ablation

Table 1 lists the patient and tumor characteristics of the index RFA, as well as the results of the RFA. The median age of patients was 55 years (range, 40–69 years), and all were male. Except for two patients without any major underlying disease, five patients (45.5%) had liver cirrhosis (one case of Child-Pugh class A, three cases of Child-Pugh class B, one case of Child-Pugh class C), three patients (27.3%) had hepatocellular carcinoma, three patients (27.3%) had synchronous head and neck cancer, one patient (9.1%) had chronic pancreatitis, and one patient (9.1%) had hypertension. Four patients (36.4%) had undergone ESD for SESCNs before index RFA. The median length of tumor area (defined as the extent of all SESCNs and/or background multiple Lugol-unstained lesions in each patient) was 16 cm (range, 6–18 cm), and the median ablation length of index RFA was 18 cm (range, 7–24 cm). Two patients (18.2%) developed esophageal strictures (defined as requiring endoscopic balloon dilatation) after index RFA, and both patients underwent ER before the index RFA. Following the index RFA, two patients required two additional circumferential RFAs due to multifocal residual SESCNs that were difficult to treat with ER.

### 3.2. Endoscopic Resection

Tumor characteristics of ER and results of ER are listed in Table 2. Out of the 17 SESCNs identified after the index RFA, eight (47.1%) were residual tumors, and nine (52.9%) were recurrent tumors. The median time between the index RFA and ER was 4.3 months (range, 2.8–5.6 months) for residual SESCNs and 13.0 months (range, 11.8–28.5 months) for recurrent SESCNs. The median tumor length was 25 mm (range, 10–66 mm) for all cases, 35 mm (range, 10–66 mm) for residual SESCNs and 23 mm (range, 18–46 mm) for recurrent SESCNs. Regarding the ER method, ESD was performed on 14 SESCNs (10 patients), and EMR was performed on three SESCNs (one patient).

The only patient who underwent three EMRs did so due to financial considerations. He had both Child-Pugh B cirrhosis and hepatocellular carcinoma and was not a candidate for esophagectomy. After the index RFA, the residual lesions were still too extensive to be treated by ER. Therefore, two additional circumferential RFAs were performed and resulted in three residual SESCNs. The first EMR session was performed for the 10 mm residual SESCN. Mucosal rupture occurred during EMR-C resulting in incomplete resection, and piecemeal resection was performed. During the second session of EMR for the residual SESCN of 21 mm in length, we performed circumferential mucosal cutting using a needle knife (KD-1L-1, Olympus, Tokyo Japan) first and then performed EMR-C. The target lesion was en bloc resected. However, esophageal stricture occurred after the second EMR-C. In the third session of EMR, the 10 mm target lesion could not be suctioned into the cap, resulting in EMR-C failure. This lesion was ablated using a hot biopsy forceps (soft coagulation 80 W, effect 4).

### 3.3. Pathological Results of the Resected Specimens

ESD resulted in en bloc resection for all target lesions. Therefore, there were l6 resected SESCNs for pathological analysis. Among them, 10 (62.5%) were HGIN and six (37.5%) were SCC.

#### 3.3.1. Neoplasia Extension to Ducts and Submucosal Glands

Neoplasia (all were HGIN) extension along the epithelial lining of ducts and submucosal glands (SMGs) was observed in eight of 16 SESCNS (50%); four out of 10 cases of HGIN (40%) and four out of six cases of SCC (66.7%). Ductal/SMG involvement was present in five out of seven residual SESCNs (71.4%) and three out of nine recurrent SESCNs (33.3%).

#### 3.3.2. Three Pathological Groups

Based on the pathological results, we divided the 16 specimens into three different groups: Group I consisted of HGIN without ductal/SMG involvement (*n* = 6, 37.5%, Figure 2b); Group II consisted of HGIN with ductal/SMG involvement (*n* = 4, 25.0%, Figure 2c); Group III consisted of SCC with muscularis mucosae or deeper involvement (*n* = 6, 37.5%, Figure 2d). These three groups may imply three possible routes in which residual or recurrent SESCN occurs.

### 3.4. Results of Post-RFA ESD Versus Primary ESD

Table 3 presents a comparison of post-RFA ESD and primary ESD. The median patient age was 53 years and 59 years (*p* = 0.064). The median tumor length was 29 mm and 35 mm (*p* = 0.222). The median ESD procedure time was 53 min and 110 min (*p* = 0.09). The median ESD procedure speed was 7.0 and 8.6 mm^2^/min (*p* = 0.644). En bloc resection rates were 100% in both groups. R0 resection rates were 85.7% (12/14) and 87.8% (65/74) (*p* = 1). Two cases with R1 resection in the post-RFA group were positive for HGIN in the lateral resection margins. Nine cases with R1 resection in the primary ESD group were positive for HGIN in the lateral resection margins (*n* = 4) and positive for SCC in the deep resection margins (*n* = 5, all were pT1b-sm2 diseases). Regarding procedure-related adverse events, delayed bleeding (defined as requiring endoscopic hemostasis) occurred in one case (7.1%) and zero cases (*p* = 0.159); esophageal stricture (defined as requiring endoscopic dilations) occurred in one case (7.1%) and 22 cases (29.7%) (*p* = 0.102). There were no cases of esophageal perforation in the two groups. In terms of depth of tumor invasion, eight (57.1%) and 28 (37.8%) tumors were located in the epithelium (*p* = 0.178); zero and 10 (13.5%) tumors infiltrated into the lamina propria (*p* = 0.353); 3 (21.4%) and 12 (16.2%) tumors infiltrated into the muscularis mucosae (*p* = 0.7); 1 (7.1%) and five (6.8%) tumors infiltrated into the superficial submucosa (within 200 μm) (*p* = 1); two (14.3%) and 19 (25.7%) tumors infiltrated into the deep submucosa (deeper than 200 μm) (*p* = 0.504).

## 4. Discussion

For balloon-type RFA, energy density settings are limited to 10 J/cm^2^ or 12 J/cm^2^ to avoid esophageal strictures. This setup ensures complete removal of the esophageal epithelium and avoids damage to the submucosa [20]. Based on this, the ER of residual or recurrent lesions after RFA may not be affected, but studies are lacking. Our findings suggest that the thermal effect of RFA can lead to mild fibrosis in the submucosa layer (Figure 3a). Submucosal fibrosis may increase the difficulty of ER, especially for EMR-C, as aspiration of the target lesion into the cap can be difficult. Therefore, we recommend using ESD instead of EMR as the method of ER after RFA. In our experience, the injection of a glycerol solution into the submucosa still adequately elevated the submucosa (Figure 3b). Submucosal dissection could be performed without difficulty, especially when we dissected along deeper submucosa planes during ESD. Therefore, the procedural speed, en bloc resection rate, and R0 resection rate of ESD after RFA were similar to those of the control group. In addition, there was no increase in adverse events related to the procedure.

The origin of residual or recurrent SESCN after RFA is unclear. According to the three pathological groups we classified in this study, we propose that there may be three possible pathways for the occurrence of residual or recurrent lesions after RFA. The first pathway is as shown by the Group I SESCNs, suggesting insufficient or incomplete contact between the electrodes of the RFA and the esophageal mucosa, resulting in incomplete ablation. Using the ablation-clean-ablation regimen, most ablated tissues would either spontaneously slough off or could be easily removed through an endoscope with a fitted cap after the first RFA pass. However, we observed in some patients that part of the ablated tissue could not be removed with the cap or even with a forceps (Figure 4). This observation is supported by an animal study showing a heterogeneous treatment effect in the TA of RFA, resulting in skipped (incompletely ablated) zones in the epithelial layer [21]. Although these incompletely ablated tissues may have the opportunity to be ablated by the secondary ablation process, some lesions may ultimately be insufficiently ablated, resulting in residual or recurrent SESCN.

The second pathway is manifested by the Group II SESCNs, which represents tumors extending beyond the ablation depth of RFA along the epithelial lining of the ducts/SMGs, resulting in incomplete treatment. This finding is consistent with other studies [17,21]. Tajima et al. reported ductal involvement in 13.8% of the 83 surgically resected SESCNs limited to the mucosa, with ductal involvement extending into the submucosa in 7.2% of the lesions [22]. Overwater et al. reported that in Paris type 0-IIb or type 0-IIc SESCNs resected by ESD, the frequency of neoplastic extension in the ducts/SMGs was 58–64% [21]. They further observed that 33% of SMGs were unaffected after balloon-based RFA. In a subsequent animal study, they found that although the overlying epithelium was ablated, 33% of SMGs remained unaffected [21]. In our study, we found ductal/SMG involvement in 50% of post-RFA ER specimens. Furthermore, we found a higher rate of ductal/SMG involvement in SCC than in HIGN (66.7% vs. 40%) and a higher rate of ductal/SMG involvement in residual SESCN than in recurrent SESCN (71.4% vs. 33.3%). These ductal/SMG lesions may constitute a hidden niche and eventually lead to residual or recurrent SESCNs [17,21]. A question for further study is how long it takes for these residual or recurrent SESCNs to occur.

The third pathway is represented by Group III SESCNs, which invade the muscularis mucosae or deeper before RFA, beyond the depth of RFA treatment. Because the ablation depth of balloon-type RFA is guaranteed in the epithelium, or at most lamina propria, there are currently no sufficiently accurate diagnostic tools to select suitable RFA candidates. Intrapapillary capillary loops type B1 lesions classified by the Japanese Esophagus Society (JES) may be the most promising candidates for RFA, but needs further studies to confirm [12,23]. Because magnifying endoscopy was not always available at our institution, we did not routinely use this modality for pre-RFA treatment assessment.

The optimal treatment options for residual or recurrent SECNs after RFA are unknown. Some studies have suggested repeating RFA for residual lesions every three months until complete remission is achieved [13,14,15,16]. However, based on our findings, only the group I SESCNs may have a chance to be cured by repeated RFA, while ESD can resect all lesions in the three groups. Incomplete RFA of SESCNs may lead to disease progression [14]. Since the available diagnostic tools cannot differentiate Group I lesions from the other two groups, we recommend that the optimal treatment for residual or recurrent SESCNs after RFA should be ESD rather than repeat RFA. With this approach, ER can compensate for the limitations of RFA, thereby minimizing the risk of incomplete treatment. This practice is critical for Group III SESCNs.

It has to be emphasized that RFA is not currently the standard treatment for SESCNs, as its long-term results have not been proven. As observed in this study, most SESCN patients underwent primary ESD rather than RFA. We performed RFA only on those patients with extensive SESCN who met strict inclusion criteria and whose lesions were potentially refractory to ESD. Most of the patients had comorbidities that were not amenable to primary surgery. For these patients, definitive radiotherapy or chemoradiotherapy may be another option. Kawamoto et al. reported that in cT1aN0M0 SESCN patients who received definitive radiotherapy or chemoradiotherapy due to being unsuitable for ER and surgery (*n* = 20), the five-year overall survival rate and disease-specific survival rate were 67% and 100%, respectively [24]. However, the local recurrence rate was 30% in their study. Additionally, grade 3 acute toxicities observed included esophagitis (10%), pneumonia (5%), and leukopenia (5%). Therefore, whether definitive radiotherapy or chemoradiotherapy is a better choice requires further studies.

This study has some limitations. Firstly, it is a retrospective study with a small number of cases. A small case number shows that the procedure is safe and feasible, but the long-term efficacy needs further research. However, since no studies have specifically reported the use of ER (or ESD) for residual or recurrent SESCNs after RFA, the results of this study may provide a basis for future studies. Secondly, extensive SESCNs often require more complex treatments. Since extensive SESCNs usually include tumor lesions of different grades, there may be sometimes multiple residual lesions after index RFA that are difficult to treat by ER. In this case, focal-type RFA may be performed before ER. In an animal study, Overwater et al. reported that focal-type RFA could ablate deeper than balloon-type RFA using the regimen of 3 × 12 J/cm^2^ [21]. They found that the ablation depth of focal-type RFA was homogenous and both the epithelium and submucosa were completely ablated. However, further studies are needed to confirm this strategy. Thirdly, although there were no recurrences in the TAs of ESD, the follow-up period was relatively short (median, 13.3 months; range, 10.4–31.3 months).

## 5. Conclusions

In conclusion, ESD for residual or recurrent SESCNs after RFA is effective and safe. Based on our findings, we suggest that ESD, rather than repeat RFA, should be the treatment of choice for residual or recurrent SESCNs after the initial RFA.

## Figures and Tables

**Figure 1 cancers-15-03558-f001:**
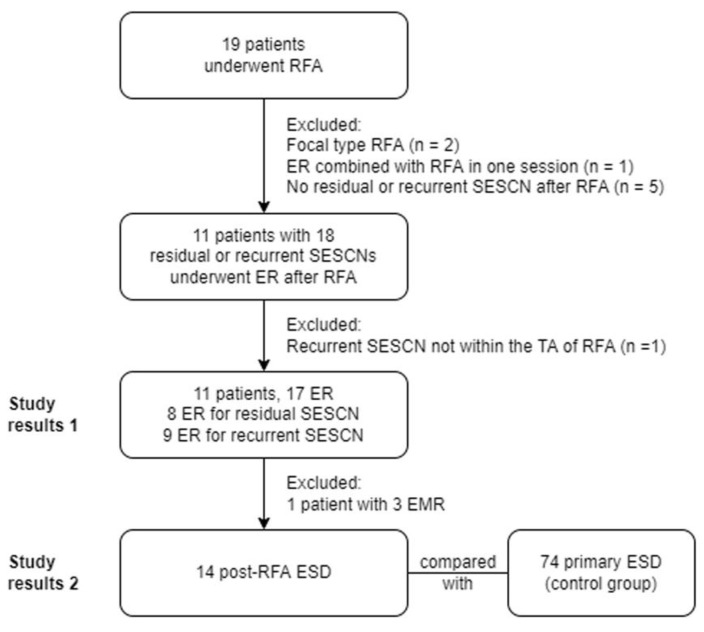
The study flowchart. Abbreviations: RFA: radiofrequency ablation; ER: endoscopic resection; SESCN: superficial esophageal squamous cell neoplasia; EMR: endoscopic mucosal resection; ESD: endoscopic submucosal dissection.

**Figure 2 cancers-15-03558-f002:**
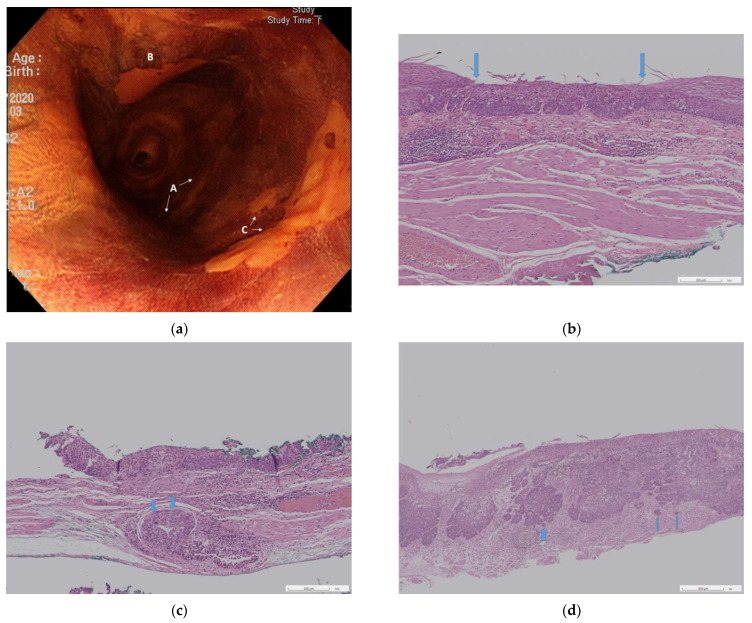
Recurrent superficial esophageal squamous cell neoplasias with variable pathology. (**a**) Endoscopy showing three recurrent lesions 11.8 months after initial radiofrequency ablation; (**b**) Pathology of lesion A in Figure 2a, showing high-grade intraepithelial neoplasia (between arrows) without ductal/submucosal glandular involvement (H&E stain, 40×); (**c**) Pathology of lesion B in Figure 2a, showing high-grade intraepithelial neoplasia with ductal/submucosal glandular involvement (arrows, H&E stain, 40×); (**d**) Pathology of lesion C in Figure 2a, showing cancer invasion to muscularis mucosae (arrows, H&E stain, 40×).

**Figure 3 cancers-15-03558-f003:**
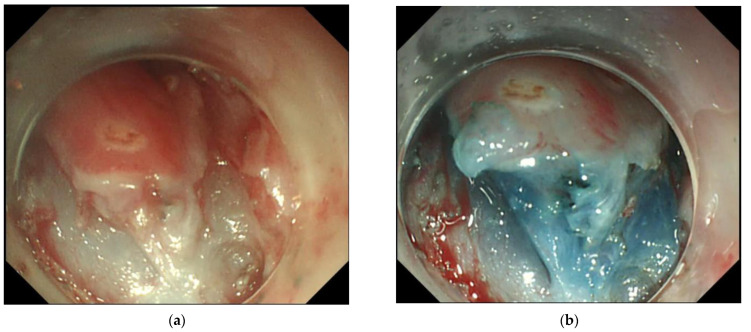
Endoscopic submucosal dissection after radiofrequency ablation. (**a**) Thermal effect of radiofrequency ablation led to mild fibrosis in the submucosa layer; (**b**) The submucosa could be sufficiently elevated by injection of a glycerol solution mixed with indigo carmine, and dissection along the deeper submucosa could be done without difficulty.

**Figure 4 cancers-15-03558-f004:**
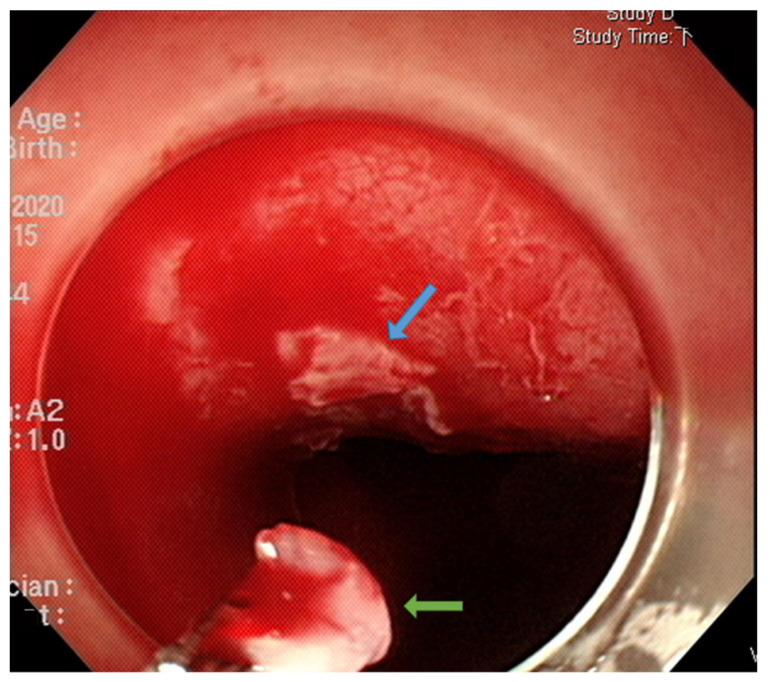
Most ablated tissues would either spontaneously slough off or could be easily removed through an endoscope with a fitted cap after the first radiofrequency ablation pass. However, some ablated tissue could not be removed due to tight adhesions (blue arrow); it was removed with a biopsy forceps (green arrow).

**Table 1 cancers-15-03558-t001:** Patient and tumor characteristics of index radiofrequency ablation.

Patient Characteristics (*n* = 11)	Number (%), or Median (Range)
Age, years	55 (40–69)
Sex, male	11 (100%)
Underlying disease	
Hypertension	1 (9.1%)
Liver cirrhosis	5 (45.5%)
Hepatocellular carcinoma	3 (27.3%)
Synchronous head and neck cancer	3 (27.3%)
Chronic pancreatitis	1 (9.1%)
Any ER before RFA	4 (36.4%)
Length of tumor area ^†^, mm	160 (60–180)
Length of index RFA, mm	180 (70–240)
Esophageal stricture after index RFA	2 ^‡^ (18.2%)
Additional RFA required after index RFA	2 (18.2%)

^†^ Length of tumor area was defined as the extent of all lesions in each patient; ^‡^ both were the patients with previous endoscopic resection; Abbreviations: ER: endoscopic resection; RFA: radiofrequency ablation.

**Table 2 cancers-15-03558-t002:** Tumor characteristics and results of endoscopic resection.

Tumor Characteristics (*n* = 17)	Number (%), or Median (Range)
Residual SESCN after index RFA	8 (47.1%)
Recurrent SESCN after index RFA	9 (52.9%)
Time between index RFA and ER, months	
For residual SESCN (*n* = 8)	4.3 (2.8–5.6)
For recurrent SESCN (*n* = 9)	13.0 (11.8–28.5)
Tumor length, mm	25 (10–66)
Of residual SESCN (*n* = 8)	35 (10–66)
Of recurrent SESCN (*n* = 9)	23 (18–46)
ER method	
ESD	14 ^†^ (82.4%)
EMR	3 ^‡^ (17.6%)
Pathologic results after ER (*n* = 16) *	
Duct/SMG involvement	8 (50%)
Of HGIN (*n* = 10)	4 (40%)
Of SCC (*n* = 6)	4 (66.7%)
Of residual SESCN (*n* = 7) *	5 (71.4%)
Of recurrent SESCN (*n* = 9)	3 (33.3%)
HGIN without duct/SMG involvement	6 (37.5%)
HGIN with duct/SMG involvement	4 (25.0%)
SCC	6 (37.5%)

^†^ In 10 patients; ^‡^ In one patient; * One residual lesion could not be resected by means of EMR; Abbreviations: SESCN: superficial esophageal squamous cell neoplasia; ER: endoscopic resection; RFA: radiofrequency ablation; ESD: endoscopic submucosal dissection; EMR: endoscopic mucosal resection; HGIN: high-grade intraepithelial neoplasia; SCC: squamous cell carcinoma; SMG: submucosal gland.

**Table 3 cancers-15-03558-t003:** Outcomes of post-RFA ESD compared with a control group of primary ESD.

	Post-RFA ESD (*n* = 14)	Primary ESD (*n* = 74)	*p*-Value
Age	53 (41–69)	59 (40–82)	0.064
Tumor length (mm)	29 (18–66)	35 (18–67)	0.222
Procedure time, min	53 (25–340)	110 (26–315)	0.09
Dissection speed, mm^2^/min	7.0 (4.4–20.6)	8.6 (2.1–54.8)	0.644
En bloc resection, *n* (%)	14 (100%)	74 (100%)	-
Completeness of resection			1
R0, *n* (%)	12 (85.7%)	65 (87.8%)	
R1, *n* (%)	2 ^†^ (14.3%)	9 ^‡^ (12.2%)	
Complications			
Significant bleeding, *n* (%)	1 (7.1%)	0	0.159
Perforation, *n* (%)	0	0	-
Esophageal stricture after ESD, *n* (%)	1 (7.1%)	22 (29.7%)	0.102
Tumor invasion depth			0.435
HIGN, *n* (%)	8 (57.1%)	28 (37.8%)	0.178
T1a-m2, *n* (%)	0	10 (13.5%)	0.353
T1a-m3, *n* (%)	3 (21.4%)	12 (16.2%)	0.700
T1b-sm1, *n* (%)	1 (7.1%)	5 (6.8%)	1
T1b-sm2, *n* (%)	2 (14.3%)	19 (25.7%)	0.504

The data of continuous variables were expressed as median (range); ^†^ Positive lateral margins (HGIN) in both cases; ^‡^ Positive lateral margins (HGIN) in four cases; positive deep margins (SCC) in five cases; Abbreviations: ESD: endoscopic submucosal dissection; RFA: radiofrequency ablation; HGIN: high-grade intraepithelial neoplasia; m2: lamina propria; m3: muscularis mucosae; sm1: tumor invasion depth within 200 μm in submucosa; sm2: tumor invasion depth over 200 μm in submucosa.

## Data Availability

Deidentified individual participant data are available and will be provided on reasonable request to the corresponding author.

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
