# Peer review of "Endoscopic Resection of Residual or Recurrent Lesions after Circumferential Radiofrequency Ablation for Flat Superficial Esophageal Squamous Cell Neoplasias"

_cancers, 2023, doi:10.3390/cancers15143558_

Round 1
Reviewer 1 Report
In this case report, the authors showed endoscopic resection of residual or recurrent lesions after circumferential radiofrequency ablation for flat superficial esophageal squamous cell neoplasias. The optimal treatment for residual or recurrent superficial esophageal squamous cell neoplasias (SESCNs) following circumferential radiofrequency ablation (RFA) is still uncertain. This study aimed to assess the effectiveness and safety of endoscopic resection (ER) for treating residual or recurrent SESCNs after RFA. Retrospective data was collected from patients who underwent RFA for SESCNs and subsequently received ER. The control group consisted of SESCN patients initially treated with primary endoscopic submucosal dissection (ESD). Eleven patients who underwent RFA had a total of 17 residual (n = 8) or recurrent (n = 9) SESCNs and were treated with ER. In one case, endoscopic mucosal resection (EMR) was unsuccessful in removing a residual SESCN. Of the 16 resected specimens, 10 were identified as high-grade intraepithelial neoplasia (HGIN) and 6 as cancer. Eight cases exhibited neoplasia that extended to the esophageal ducts or submucosal glands (SMGs). Pathological results suggested three possible routes for the occurrence of residual or recurrent SESCNs: HGIN without ductal or SMG involvement (37.5%), HGIN with ductal or SMG involvement (25.0%), and squamous cell carcinoma (SCC) with involvement of the muscularis mucosae or deeper layers (37.5%). Compared with the control group, the study group showed similar procedural speed, en bloc resection rate, R0 resection rate, and complication rate. In conclusion, post-RFA ER demonstrated comparable safety and efficacy to primary ESD. Therefore, ESD should be considered the preferred treatment for residual or recurrent SESCNs following initial RFA. Overall, this case report is interesting and may be helpful for the clinical trial study. I only have some minor concerns that need to be addressed by the authors.
1: The authors should provide or discuss the best treatment methods after RFA.
2: The authors mentioned that the study group had similar procedural speed, en bloc resection rate, R0 resection rate, and complication rate compared to the control group. Also, the safety and efficacy of post-RFA ESD were similar to those of primary ESD. If like this, why ESD should be the treatment of choice for residual/recurrent SESCNs after initial RFA?
3: Word usage need to be polished.
Minor editing of the English language required
Author Response
Dear Reviewer:
Thank you for reviewing our manuscript and for your excellent suggestions, which will greatly improve our manuscript. Here is our reply:
- We have revised the discussion on optimal treatment options for residual or recurrent SECN after RFA in the Discussion section (Lines 298-308). In addition, we have added a discussion that definitive radiotherapy or chemoradiotherapy may be an alternative option for patients with extensive SESCN who are not candidates for endoscopic resection or primary surgery (Lines 309–321).
- We have revised this discussion in the Discussion section (Lines 259-308). Based on our findings, only the group I SESCNs (37.5% of all SESCNs) may have a chance to be cured by repeated RFA. The lesions in Group II SESCN (25%) and Group III SESCN (37.5%) exceeded the ablation depth of RFA, resulting in incomplete RFA treatment. However, ESD can cure lesions in all three groups. Therefore, we recommend that the optimal treatment for residual or recurrent SESCNs after RFA should be ESD rather than repeat RFA.
- As shown in the revised manuscript, the English language has been improved.

Reviewer 2 Report
Small numbers indicate that this procedure is safe and feasible and long term efficacy uncertain
Author Response
Dear Reviewer:
Thank you for reviewing our manuscript and for your excellent suggestions. We believe that your suggestions will greatly improve our manuscript. Here is our reply:
- We have added your comment in the study limitation section of the Discussion (Lines 324-325).
Reviewer 3 Report
This is a small retrospective study of 11 patients who underwent endoscopic resection (ER) after endoscopic RFA of extensive esophageal superficial SCC(ESCC). There are a number of concerns in the current study:
1. Although RFA has been reported in some small scale studies, the long term outcome in esophageal SCC has not been proven and it should be emphasised that it is not a standard therapy for ESCC.
2. The recurrence / residual lesion rate in the RFA patients was much higher than that reported from previous studies (only 5 out of 19 patients did not develop recurrence / residual). What was the reason for the high residual / recurrence rate? Was the indication of RFA different from the previous studies?
3. The authors mentioned that the indication of RFA includes (1) flat ESCC >3/4 circumference >5cm or (2) multiple lugol's voiding lesions. In the cohort, how many of the patients belong to second group? Alternatively, how many patients presented with nodular lesions at the beginning?
4. Can the authors comment if radical chemoRT / radical RT could be another option for these patients with primary ESCC. Judging from the multiple issues raised by the authors in the discussion part, would radical RT be a better alternative to eradicate the early cancers while ER be saved by local recurrence?
5. It could be useful to describe the endoscopic findings of residual / recurrent lesions after RFA, especially in relation to whether it could predict the presence of ductal involvement / submucosal gland involvement.
Author Response
Thank you for reviewing our manuscript and for your excellent suggestions, which will greatly improve our manuscript. Here is our reply:
- We have added your suggestion to the discussion section to emphasize that RFA is not currently the standard treatment for SESCNs (Lines 311-316).
- Yes, the indications for RFA in this study were different from previous studies. Unlike others, we only enrolled patients with extensive SESCN refractory to endoscopic resection. For extensive SECN, there are often different grades of tumor invasion. Therefore, there will be an increased risk of including the lesion beyond the depth of RFA treatment. Furthermore, the more extensive the lesion, the higher the risk of incomplete RFA treatment. These two are the reasons for the higher incidence of incomplete RFA than in other studies.
- Extensive SESCNs in this study include (1) at least one or more SESCNs invading >3/4 of the esophageal circumference and greater than 5 centimeters in length (n = 6), or (2) multiple Lugol-unstained lesions, with/without connections, ranging from > 3/4 esophageal circumference and length greater than 5 cm (n = 5). (Lines 74-77). As we included only flat lesions (defined as Paris classification type 0-IIb or 0-IIb+0-IIc), no patient presented with nodular lesions in our study. (Lines 82-83).
- We have added that definitive radiotherapy or chemoradiotherapy may be an alternative treatment option in the Discussion section (Lines 316-323).
- Currently, the available endoscopic modalities cannot distinguish between group I lesions (HGIN without ductal/SMG involvement) from group II lesions (HGIN with ductal/SMG involvement) and group III lesions (cancer involving the m3 layer or deeper). This is why we suggested that the best treatment for residual or recurrent SECN after RFA should be ESD rather than repeated RFA because ESD can resect all lesions in the three groups (Lines 301-310).

Round 2
Reviewer 3 Report
Thank you for the revised manuscript. I have no further comments.